# A Highly Sensitive and Miniature Optical Fiber Sensor for Electromagnetic Pulse Fields

**DOI:** 10.3390/s21238137

**Published:** 2021-12-06

**Authors:** Min Zhao, Xing Zhou, Yazhou Chen

**Affiliations:** National Key Laboratory on Electromagnetic Environment Effects, Army Engineering University, No. 97 Heping West Road, Shijiazhuang 050003, China; zhao_min_2012@126.com (M.Z.); zxlwbh@126.com (X.Z.)

**Keywords:** EMP measurement, optical fiber sensor, direct electro-optic conversion, linear response, time domain, frequency domain

## Abstract

The detection of an electromagnetic pulse (EMP) field is of great significance in determining the field environment of tested equipment in small spaces. Finger-shaped miniature optical fiber sensors for electromagnetic pulse field measurement were designed. The antenna of a weak field sensor was integrated with a shielding shell, and the wire welded at the direct electro-optic converting circuit connected to an optical fiber through special structure and circuit design was taken as the antenna of a strong field sensor. Measurements in the time domain and frequency domain had been carried out for the two sensors. Experiment results demonstrate that the weak field sensor and the strong field sensor have flat responses from 100 kHz to 1 GHz with a variation of 2.3 dB and 2.9 dB, respectively, and the EMP waveform detected by the sensors agrees well with the applied standard square wave. Moreover, the strong field sensor exhibits linear responses from 645 V/m to 83 kV/m. The resolution of the weak field sensor is as low as 13 V/m. The result indicated that the designed sensors had good performance.

## 1. Introduction

The measurement of an electromagnetic pulse (EMP) field is of great importance in an immunity test of electromagnetic compatibility, especially in small spaces such as in vehicles, aircraft and ship cabins. Currently, there are mainly two methods for realizing EMP field measurements. One method utilizes a broadband antenna to take measurements directly for which its typical application includes D-dot sensors [1,2]. These kinds of sensors can cause severe distortions in tested EMP fields in small spaces because of its large volume, as well as the metallic structures of the antenna and coaxial cables used to transmit electric field signals. Another method is called electro-optic modulation, which includes active electro-optic modulation and passive electro-optic modulation. This method adopts optical fibers for transmitting optical signals converted by the electro-optic modulation circuit, which eliminates the influence of a coaxial cable [3,4,5,6,7,8,9,10]. For active electro-optic modulation, i.e., the method adopted by this article, the electrically smaller antenna was utilized to induce external EMP; generally, an electric field test uses an electric dipole antenna, and the magnetic field test uses a magnetic small ring antenna. Then, a laser diode (LD) or a light emitting diode (LED) acts as the optical component, which is used to convert the EMP field signal into an optical signal, which is then demodulated after transmission by an optical fiber, and the measured EMP field signal will be restored and observed by the oscilloscope. In addition, the integrated optical wave-guide sensor is a kind of typical passive electro-optic modulation that uses three Lithium niobate (LiNBO3) and has the main advantage of high field testing, possessing a miniature size and having low influences on the tested field. However, some defects still exist, such as high fabrication cost and low sensitivity [11,12,13,14,15,16,17]. Furthermore, the size and measurement time of the sensors based on active electro-optic modulation are limited since they are powered by batteries [18,19,20,21]. Currently, researchers are mainly focused on decreasing its size and field interference in order to enhance bandwidth, sensitivity and accuracy [22,23,24,25,26,27].

In this article, a finger-shape miniature optical fiber sensor using LD and special structural design was presented for EMP field measurement, which could overcome the problem of large volume of active electro-optic sensors and increase its sensitivity. The frequency response of the sensors was measured with a Gigahertz transverse electromagnetic (GTEM) cell (100 kHz–450 MHz) and conducted in a microwave chamber (450 MHz–1 GHz). Moreover, the time domain and linear characteristics of the sensors are tested by square wave pulse and double exponential pulse, where the former is generated by a high frequency noise simulator and GTEM and the latter is produced by a double exponential pulse source and parallel plate transmission line. Furthermore, the analytic model, the simulation analysis, the core circuit and configuration of the sensor are reported in detail.

## 2. Operational Principles

### 2.1. Sensor Design

The configuration of the proposed sensor system is shown schematically in Figure 1. The sensor shell acts not only as the dipole antenna detecting an externally weak EMP field but also shields the battery and the electro-optic modulation circuit shown in Figure 2.

The entire shell is divided into left and right parts by an insulating medium: The left shell is equivalent to a grounding mechanism, and the right shell plays a different role according to the strength of the EMP field. For the weak field sensor, the right shell is electrically connected with the antenna port of the electro-optic modulation circuit by a wire; in this case, the length of the right shell is similar to length of the left shell, and this kind of shell structure acts as a dipole antenna for detecting small signals. However, when the detected signal is so large that it exceeds the linear range of the FET Q1 and enters into the saturation region of FET Q1, it can cause a distortion of the detected signal. Obviously, the above dipole antenna is not suitable for detecting a strong EMP field. Therefore, the strong field sensor was designed by changing the type of the antenna, and we took a short wire welded at the antenna port of the electro-optic modulation circuit as the mono-pole antenna; here, the right shell is only a shielding shell and not the antenna, and the left shell is still taken as the ground. The two kinds of antenna both belong to an electrically small antenna, which has high resonant frequency and is connected to the grid electrode of the FET Q1 through a attenuation capacitor C_L_. Moreover, the LD U1 is connected in series with the FET Q1. Due to the fact that the FET is a voltage-controlled current element and has high input resistance, the electric field signal induced by the antenna directly controls the drain current of the FET and realizes the high resistance output of the antenna-received signal, which is the antenna’s high resistance coupling principle of the electric field sensor. According to the analysis theory of the antenna, when the antenna is coupled with high resistance, the operating bandwidth of the antenna is fully utilized, which provides a broadband foundation for a pulse electric field test. The LD is connected in series with the FET; thus, the drain current of the FET becomes the operating current of the semiconductor laser, which realizes the direct drive of the LD by the antenna receiving the electric field signal, and finally achieves a direct conversion between the electric field signal and the modulated optical signal, which widens the bandwidth of the test system from the source. Then, the transmitted optical signal is received by the receiver and converted into an electrical signal, which is displayed by the oscilloscope.

For the proposed direct electro-optic modulation circuit, the amplification, conversion and drive are achieved by only one FET, and the circuit is simplified further on the basis of the literature [26]. Moreover, this design integrated the antenna and electro-optic modulation together and eliminates the impedance conversion network and amplification-driving circuit that seriously limits the transmission bandwidth; this fundamentally overcomes the problem of the connection between the electro-optic modulator and the antenna. In the electro-optic modulation circuit, the FET adopted an N-channel GaAs MESFET NE72218 produced by NEC company, and Table 1 presents its electrical characteristics. It can be observed from the Table 1 that it possesses low phase noise and high power gain, its saturated drain current reaches 120 mA and the cut-off upper frequency reaches up to 12 GHz. Figure 3b shows the relationship curve between drain current I_D_ and drain-to-source voltage V_DS_ under different grid-to-source voltage V_GS_; here, V_GS_ represents the voltage of the input signal received by antenna, and I_D_ is used to drive LD. According to the parameters of FET, LD is customized by Jiuzhou company, which is a high-performance uncooled distributed feedback semiconductor laser (DFB–LD) with the wavelength of 1310 nm, an output power of 7 mW and a cut-off frequency of 2.5 GHz; moreover, the physical diagram of the laser is shown in Figure 4a,b, and the figures show that the laser has a threshold current *I*_t_, which is 4.5 mA. When the driving current generated by FET is less than the threshold current, the laser basically does not emit light or only emits very weak spectral lines. Conversely, when the driving current is greater than FET, the laser starts to emit a laser, and the output light intensity increases linearly with the increase in driving current, which is the operating area of electro-optic modulation. The linear region consistency of the two components determines the performance of the sensor.
(1)$$U_{GSQ}=\frac{R_{1}}{R_{1}+R_{2}}\times V_{GG}-I_{DQ}\times R_{S}$$
(2)$$U_{DSQ}=V_{DD}-I_{DQ}\times (R_{S}+R_{d})$$

Therefore, it is essential to set an appropriate static operating point for FET in order to ensure an input electrical signal that works in a linear region. Figure 3a shows the DC path of electro-optic modulation circuit. The FET Q1 is operated in terms of self bias, and the resistance, Rs, can adjust the self bias of the FET and ensure that the FET is in a suitable linear operating area and set a suitable operating current for the LD. Moreover, a current negative feedback circuit through Rs is presented, which provides a stable operating current for the LD. By conducting several tests, the optimal values of these parameters were determined, which include the following: V_GG_ = −1.24 V; V_DD_ = 3.3 V; Rs = 11 Ω; R_1_ = 68 MΩ; R_2_ = 43 MΩ; R_d_ = 3.3 Ω; I_DQ_ = 28 mA; and U_GSQ_ and U_DSQ_ are calculated by Equations (1) and (2) and equal −1.07 V and 2.1 V, respectively. The static operating point can be observed in Figure 3b.

The measurement range of the sensor is mainly determined by the threshold current *I*_t_ of LD and the saturated drain current I_DSS_ of the FET. *I*_t_ corresponds to the minimum electric field strength and determines that FET is always in the amplification region. Due to the fact that the output power of LD has a large transient allowable value, I_DSS_ represents the maximum electric field strength. By taking the static operating point Q of the FET as a reference point, when the input voltage in the grid electrode of FET was changed from −0.635 V to +0.635 V (that is, the AC signal is superimposed on the basis of U_GSQ_), the maximum and minimum grid-to-source voltages are −0.81 V and −2.08 V, respectively; at the same time, the maximum and minimum drain currents are 4.5 mA and 120 mA, respectively. Hence, the corresponding output optical power range of LD is 0–7 mW. In this case, the sensor can measure negative and positive pulses in an undistorted manner. For the case of only negative pulse or positive pulse, U_GSQ_ can be adjusted by R_1_, R_2,_ Rs and V_GG_ to increase or decrease, and the measurement range of 0–1.27 V can be achieved when U_GSQ_ is assumed at −2.08 V. Similarly, when U_GSQ_ is set at −0.81 V, the negative pulse with voltage varying from −1.27 V to 0 V could be measured. In addition, when the input voltage is greater than the above measurement range, the appropriate attenuation capacitor C_L_ is needed, which can be further broaden the measurement range, and the factor depends on the equivalent capacitance of antenna *C*_ant_ and its capacitance value.

### 2.2. Analytic Model

In order to analyze the influence of various parameters on the impedance of antenna and its terminal load comprehensively, the equivalent circuit of an electric field sensor was established, as shown in Figure 5. *C*_ant_, *R*_ant_ and *L*_ant_ are the equivalent capacitance, resistance and inductance of antenna, respectively. *R*_L_ and *C*_L_ are the resistance and capacitance of loads.

The voltage of load *V*_L_ is calculated in the *S* domain and expressed as transfer function G(*s*).
(3)$$G(s)=\frac{V_{L}(s)}{V_{o}(s)}=\frac{Z_{L}(s)}{Z_{ant}(s)+Z_{L}(s)}=\frac{sC_{ant}R_{L}}{\left(1+sR_{L}C_{L})(1+sR_{ant}C_{ant}+s^{2}L_{ant}C_{ant}+\frac{sC_{ant}R_{L}}{1+sR_{L}C_{L}}\right)}$$

In the low frequency band, the second-order pole in Equation (3) is far greater than (1 + *sR*_L_*C*_L_) (that is, *s*^2^*L*_ant_*C*_ant_ << 1, *sR*_ant_*C*_ant_ << 1), and Equation (3) can be simplified as follows.
(4)$$G(s)=\frac{sC_{ant}R_{L}}{1+sR_{L}(C_{ant}+C_{L})}$$

The above equation is regarded as a high-pass filter for which its lower cut-off frequency *f*_L_ can be written as follows.
(5)$$f_{L}=\frac{1}{2\pi R_{L}(C_{ant}+C_{L})}$$

In the high frequency band, Equation (3) can be simplified as a second-order low-pass filter:(6)$$G(s)=\frac{C_{ant}}{C_{ant}+C_{L}}\cdot \frac{\omega_{0}^{2}}{s^{2}+2\zeta \omega_{0}s+\omega_{0}^{2}}$$
where ω_0_ = ((*C*_ant_ + *C*_L_)/(*C*_L_*C*_ant_
*L*_ant_))^1/2^ is the undamped angular frequency of the second-order system, and ζ = *R*_ant_/(2*L*_ant_ω_0_) is the damping ratio. Suppose *s* = *jw*, *u* = *w*/*w*_0_; then, Equation (6) can be written as follows.
(7)$$G(u)=\frac{C_{ant}}{C_{ant}+C_{L}}\cdot \frac{1}{1+j2u\zeta -u^{2}}$$

The amplitude and phase characteristics are given, respectively, as follows.
(8)$$\left\{ \begin{array}{l} \left|G(u)\right|=\frac{C_{ant}}{C_{ant}+C_{L}}\cdot \frac{1}{\sqrt{{\left(1-u^{2}\right)}^{2}+{\left(2u\zeta \right)}^{2}}}\\ ∠G(u)=-\tan^{-1}\frac{2\xi u}{1-u^{2}} \end{array}\right.$$

According to Equation (8), the high-frequency response characteristic curve is drawn in Figure 6. When 0 < ζ < 0.707, the upper cut-off frequency *f*_H_ can be expanded, but the resonance spike in the curve will appear, and its peak value *M*_r_ increases with a decrease in ζ. By setting *d*/*du*|G(*u*)| as 0, the resonant frequencies *f*_r_ and *M*_r_ can be derived as follows [28].
(9)$$f_{r}=\frac{\omega_{0}}{2\pi }\sqrt{1-2\zeta^{2}}$$
(10)$$M_{r}=\frac{C_{ant}}{C_{ant}+C_{L}}\cdot \frac{1}{2\zeta \sqrt{1-\zeta^{2}}}$$

According to the definition of −3 dB bandwidth, suppose |G(*u*)| = 0.707 × *C*_ant_/(*C*_ant_ + *C*_L_); then, its upper cut-off frequency *f*_H_ can be described as follows.
(11)$$f_{H}=\frac{\omega_{0}}{2\pi }\sqrt{(1-2\zeta^{2})+\sqrt{4\zeta^{4}-4\zeta^{2}+2}}$$

It can be observed from Figure 6 that there is no resonance spike in the curve, and the curve rolls down rapidly at the upper cut-off frequency when ζ is greater than or equals 0.707. In particular, when ζ equals 0.707, the flat section of the amplitude frequency characteristic curve is the widest, and the phase frequency characteristic curve is close to an oblique line; thus, it is usually called the best damping ratio. In this case, *f*_H_ equals *f*_0_ according to Equation (11); thus, its upper cut-off frequency *f*_H_ can be simplified into the following.
(12)$$f_{H}=\frac{1}{2\pi }\sqrt{\frac{1}{L_{ant}}(\frac{1}{C_{ant}}+\frac{1}{C_{L}})}$$

When the frequency *f* in the transient electric field to be measured is between *f*_L_ and *f*_H_, the time domain output of antenna *V*_L_(*t*) on the terminal load can be written as follows:(13)$$V_{L}(t)=\frac{C_{ant}}{C_{ant}+C_{L}}V_{o}(t)$$
where C_ant_ = h/(60 × *c* × (2 × ln(2 × h/a) − 2 − ln4)), *h* is the length of antenna, *a* is the radius of antenna and c is the propagation velocity of electromagnetic wave.

### 2.3. Simulation Nalysis

By taking the cylindrical electrically small dipole antenna as an example, the influence of the parameters such as the size of the antenna and its termination load on the frequency response of the sensor is further studied. The input impedance *Z*_ant_ of the antenna is expressed as follows:(14)$$\{\begin{array}{c} Z_{ant}(\omega )=R_{ant}(\omega )+jX_{ant}(\omega )\\ R_{ant}(\omega )=\frac{\sinh2h\eta -\frac{\eta }{k}\sin2hk}{\cosh2h\eta -\cos2hk}Z_{0}\\ X_{ant}(\omega )=-\frac{\sin2hk+\frac{\eta }{k}\sinh2h\eta }{\cosh2h\eta -\cos2hk}Z_{0} \end{array}$$
where *Z*_0_ = 120 × (ln(2 × *h*/*a*) − 1), which is the average characteristic impedance of the antenna; *η* = 73.1 × (*Z*_0_ × *h* × (1 − sin(2 × *h* × *k*)/(2 × *h* × *k*))), which is the attenuation constant of the equivalent transmission line after considering radiation loss; and *k* = *ω*/*c*, which is the wave number.

According to the equivalent circuit model in Figure 2, the relationship between *V*_L_ and *V*_o_ is written in the form of a transfer function, and its frequency domain expression is described as follows:(15)$$G(\omega )=\frac{V_{L}(\omega )}{V_{o}(\omega )}=\frac{Z_{L}(\omega )}{Z_{ant}(\omega )+Z_{L}(\omega )}$$
where *Z_L_*(*ω*) = R*_L_*/(1 + *jω* × *R_L_* × *C_L_*).

The simulation results of frequency and time domain characteristic of the sensor under the different size of antenna and load resistance are given in Figure 7, Figure 8 and Figure 9. The length and radius of the antenna are fixed at 30 mm and 3 mm, respectively, changing the value of *R*_L_, i.e., 50 Ω, 1 kΩ and 1 MΩ, and *C*_L_, i.e., 1 pF, 10 pF and 50 pF. The frequency characteristics of the sensor were analyzed based on Equations (14) and (15), which can be observed from Figure 7a.

It can be observed from Figure 7a that the change of *R*_L_ and *C*_L_ can obviously affect the low-frequency characteristic of the sensor, but it has little influence on its high-frequency characteristic. By fixing *R*_L_ at 1 MΩ and increasing *C*_L_ from 1 pF to 50 pF by a certain step, the lower cut-off frequency moves towards the direction of low-frequency; this trend broadens the bandwidth of sensor, but the cost is a decrease in gain. Conversely, by fixing *C*_L_ at 50 pF and increasing *R*_L_ from 50 Ω to 1 MΩ by a certain step, in this case, the change trend of the lower cut-off frequency is similar to the above case, and the only difference is that the gain does not decrease but retains a fixed value, which agrees with the theoretical calculation of the sensor’s lower-frequency in Section 2.2. Furthermore, assuming that *R*_L_ is 1 MΩ and *C*_L_ is 10 pF, when the size of antenna is changed, the effect on the frequency characteristic of the sensor can be observed from Figure 7b, and the result shows that the change of the antenna’s length and radius generates a distinct influence on the upper cut-off frequency of the sensor but has little effect on its lower cut-off frequency. Furthermore, when the radius *a* of antenna is maintained at 50 mm, increasing the length *h* of antenna causes the gain of the sensor to also increase, but the upper cut-off frequency obviously decreases. On the other hand, when the length *h* of antenna is fixed at 1 mm, increasing the radius *a* of the antenna causes the gain of the sensor to also increase, but its lower and upper cut-off frequencies do not experience significant change. Meanwhile, compared with the resonance spike of these curves in Figure 7b, it can be observed that the resonance peak of the curve becomes more obvious with an increase in *h*; that is, the damping ratio ζ shows a decreasing trend. However, with an increase in *a*, the curve of the frequency response tends to be flat, and the damping ratio ζ has an increasing trend, which means that the physical size of antenna can affect damping ratio ζ. Therefore, by optimizing the design of antenna, the frequency response characteristic curves with maximum flatness can be obtained.

For the dipole antenna as an example, the simulation model of the sensor was established (as observed in Figure 8a). The material of the antenna is set as PEC, i.e., ideal conductor material, which is placed 1 m away from the incident plane wave, and the termination load is a circuit of *R*_L_ in parallel with *C*_L_. Considering that the square pulse with a sharp edge has rich low frequency and high frequency components (that is, its response can directly reflect the performance of the sensor), the square pulse with 1 kV of the voltage peak, 0.5 ns of rise time and 20 ns of the pulse width was selected as the excitation signal (see Figure 8b), which can form a plane wave environment with 1 kV/m of EMP field strength at the position where the sensor was placed. In this case, Figure 9 shows the pulse responses of loads under different load resistances when *h* is 30 mm and *a* is 3 mm and the different sizes of the antenna when *R*_L_ is 1 MΩ and *C*_L_ is 10 pF.

It can be observed from Figure 9a that the response waveform of the sensor incurs obvious flattop declines when *R*_L_ is 50 Ω and *C*_L_ is 50 pF, which is similar to the differential waveform of square pulses. This case demonstrates that the sensor possesses bad responses characteristic of lower cut-off frequencies. With an increase in *R*_L_, the response waveform becomes better, and the excitation signal can be restored correctly, except for the attenuation on magnitude when *R*_L_ is 1 MΩ. In this case, by increasing capacitance *C*_L,_ its magnitude will decrease gradually. As a result, a greater *R*_L_ can reduce the lower cut-off frequency of the sensor effectively, and a lower *C*_L_ can improve its gain (see Figure 9b). Moreover, Figure 9c shows that the gain of the sensor will obviously increase with an increase in the antenna’s length *h* or radius *a*. Furthermore, when *a* is fixed to 1mm and the length *h* of antenna is increased, a larger vibration occurs at the positions of the rising and falling edge. However, as shown in Figure 9d, the vibration of the edge cannot be improved by changing radius *a.* Therefore, the degree of edge’s vibration, i.e., the higher upper cut-off frequency of sensor, mainly depends on length *h* of antenna.

### 2.4. Selection of Key Parameters

On the basis of the above analysis, *f*_L_ can be reduced by increasing *R*_L_, *C*_L_ and *C*_ant_. The increase in *C*_ant_ results in a decrease in its resonant frequency, which limits the high-frequency response characteristics of the sensor. Generally, in order to expand *f*_H_, C_ant_ was designed to be relatively small (several pF). The value of *C*_L_ mainly depends on input capacitance and parasitic capacitance of the load circuit. It can be observed from simulation results that an increase in *C*_L_ reduces the output gain and affects the high frequency performance of the sensor. Therefore, the method of increasing *R*_L_ is used to reduce *f*_L_. If we need to satisfy the condition that *f*_L_ is less than 100 kHz, where *C*_L_ + *C*_ant_ equals to 100 pF approximately, then the value of *R*_L_ must be greater than 10^6^ Ω.

It can be observed from Equation (12) that *f*_H_ is greatly affected by *C*_ant_, *L*_ant_ and *C*_L_. Reducing these parameters can improve *f*_H_. This shows that the physical characteristics of the antenna itself such as its length *h* and radius *a* have a great influence on the high-frequency response characteristics of the electric field sensor. According to the theory of the antenna, the maximum upper cut-off frequency *f*_hm_ of the sensor can be satisfied as follows.
(16)$$f_{hm}\ll \frac{c}{h}$$

Therefore, if *f*_H_ is required to reach 1 GHz, then *h* should be less than 60 mm. Moreover, in order to decrease the influence of the parasitic parameter, a miniature circuit design is adopted for the PCB layout of the electro-optic modulation circuit shown in Figure 10a. Figure 10b,c show the photographs of two sensors. For the weak field sensor, the right part of the shielding shell acts as an antenna (see Figure 10b). Its package size is *Φ* 13 mm × 70 mm, *h* is 35 mm, *a* is 6.5 mm and *R*_L_ equals R1/R2, which is 26 MΩ. For the strong field sensor, the wire acts as an antenna (see Figure 10c). Its length can be easily adjusted. Here, the package size is Φ 13 mm × 60 mm, *h* is 3 mm, *a* is 0.5 mm, *C*_L_ is 100 pF and *R*_L_ is also 26 MΩ. The designed strong field sensor can measure higher fields and has a wider measurement range. Conversely, the designed weak field sensor tends to improve sensitivity at the cost of reducing the measurement range, which can be used to test the very low field strength by removing attenuation capacitance C_L_. Due to the fact that these circuits use less dissipative elements, a 3.7 V battery can be guaranteed for the normal operation of the circuit for several hours.

## 3. Experimental Results

### 3.1. Frequency Domain

Sensors’ bandwidth and flatness can be obtained by a frequency domain test [29,30]. Due to the existence of high-order modes in the GTEM chamber, the chamber is not suitable for sensor calibration at high frequencies. Referring to IEEE 1309-2005, the frequency response of the sensor is characterized in a GTEM cell (100 kHz–450 MHz) and in a microwave chamber (450 MHz–1 GHz). The frequency domain test configuration diagrams based on a GTEM cell and microwave chamber are shown in Figure 11a,b, respectively. The signal source and power amplifier generate a plane wave environment in the GTEM cell or microwave chamber, which is measured by continuous wave field intensity meter (i.e., EMR 200) and the sensor at the same time. The EMR 200 and sensor are placed in parallel under a uniform field region (see Figure 11c), and we take EMR 200 as the standard sensor and adjust the output power of the signal source in order to retain the field strength in the GTEM cell and microwave chamber measured by EMR 200 at a fixed value (i.e., 50 V/m for the strong field sensor and 30 V/m for the weak field sensor); at the same time, the corresponding output power *P* of the sensor system at a certain frequency can be observed by the spectrum analyzer. Thus, the corresponding relationship between the output power of the sensor and the field strength of EMR 200 can be obtained. Then, we alter the frequency from 100 kHz to 1 GHz at a certain step and repeat the above operation; in this case, the curve of frequency response can be drawn.

The frequency responses of the tested sensors are shown in Figure 12. The variation of the frequency response of sensor Δ*P* within a certain frequency range is defined as follows.
(17)$$\Delta P=\left|P_{\max}-P_{\min}\right|$$

Generally, when Δ*P* is less than 3 dB, we take the frequency range as the frequency bandwidth of the sensor. It can be observed from Figure 12 that the *P*_max_ of the strong field sensor is −43.6 dBm at 800 MHz, and *P*_min_ is −40.7 dBm at 900 MHz. Δ*P* equals 2.9 dB. For the weak field sensor, *P*_max_ and *P*_min_ are −22.8 dBm at 100 kHz and −25.1 dBm at 200 MHz, respectively, and Δ*P* equals 2.3 dB. Obviously, Δ*Ps* of the two sensors are both less than 3 dB; thus, the frequency bandwidth from 100 kHz to 1 GHz of them can be determined.

### 3.2. Time Domain

In order to determine the time domain performance of the proposed sensor, we adopted the standard field method [31,32,33]. Since the range of electric field strength in the reference field required for calibration is wide, a set of pulse sources cannot meet the test in the entire range with respect to accuracy and range. A division into a weak field strength test and strong field strength test is required, respectively. For the weak field strength test, the square wave pulse field is used as the reference field. The square wave pulse field generation device is composed of a square wave source with an amplitude range of 0–4 kV, rise time of 1 ns and pulse width of 10 ns–1000 ns and a GTEM cell. The test configuration is shown in Figure 13a. In the uniform field area, the square wave pulse fields with different field strengths can be acquired by properly adjusting the position of the sensor between the core plate and the bottom of the GTEM cell. The standard square wave field strengths vary from 13 V/m to 13,225 V/m. For the strong field strength test, the double exponential pulse field is used as the reference field. The test configuration is shown in Figure 13b. A double exponential pulse field is generated in the parallel plate transmission line, and the double exponential pulse source has a rising time of 2 ns, a pulse width of 25 ns and an amplitude of 50 kV. The height of the parallel section is 0.6 m; thus, the pulse field intensity generated in the parallel plate transmission line can output a field strength of about 5000 V/m–83,333 V/m. The sensor is located in the parallel section and is connected to the optical receiver by the optical fiber. Simultaneously, an oscilloscope was utilized to observe the voltage pulse waveform generated by the optical receiver, and the measured field strength can be converted according to the voltage.

The detected waveform in the time domain for the strong field sensor is shown in Figure 14 when the applied electric field measured 8366 V/m. It can be observed from Figure 14 that the response of the sensor agrees well with the applied electric field in terms of rise time and pulse width.

Figure 15 shows that the relation between electric field *E* and output voltage *V*_out_ detected by the oscilloscope. As observed in Figure 15b, the measurable linear electric field ranges from 645 V/m to 83 kV/m for the strong field sensor. The converted coefficient *k* from the electric field-to-output voltage is approximately 94 by examining the linear fit curve, and the correlation coefficient of the sensor is 99.98%, which indicates that the sensor possesses good linearity. With regard to the measurable linear electric field range of sensor, it mainly depends on the linear range of FET adopted; the position of the linear area can be adjusted according to the polarity of the test waveform. In any case, once the output voltage exceeds the maximum value of the linear range, it will no longer increase linearly. Moreover, due to the fact that the relation between the output voltage and the electric field is related to many factors, such as antenna parameters, attenuation capacitor C_L_, magnification of FET, the receivers and so on, it is hard to obtain the electric field by calculation; it can only be obtained by conducting tests. As observed from Figure 15a, when the measured field is 83 kV/m, the corresponding voltage value is only 0.8 V, which does not reach the saturation voltage of 1.3 V of FET. Therefore, we can obtain a measurable field greater than 83 kV/m. In addition, the designed weak field sensor has low energy consumption and is powered by a battery; thus, the base noise of sensor system is only 3 mV, which is known by conducting the test. For the weak field sensor, the minimum detectable electric field in the time domain can be tested at approximately 13 V/m (see Figure 15a).

## 4. Conclusions

A finger-shaped miniature optical fiber sensor was presented by using LD and a special structural design for EMP field measurement. Experimental results show that the frequency response of the sensor is flat from 100 kHz to 1 GHz. The detected electric field waveform in the time domain agrees well with the applied standard square wave. In addition, the measurable linear electric field range ranged from 645 V/m to 83 kV/m, while the minimum detectable electric field in the time domain was tested at approximately 13 V/m. In conclusion, the sensor system has good performance for pulsed electric field measurements.

## Figures and Tables

**Figure 1 sensors-21-08137-f001:**
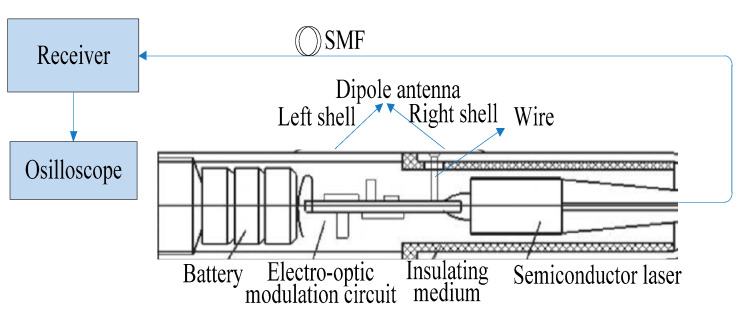
Configuration of the proposed sensor system.

**Figure 2 sensors-21-08137-f002:**
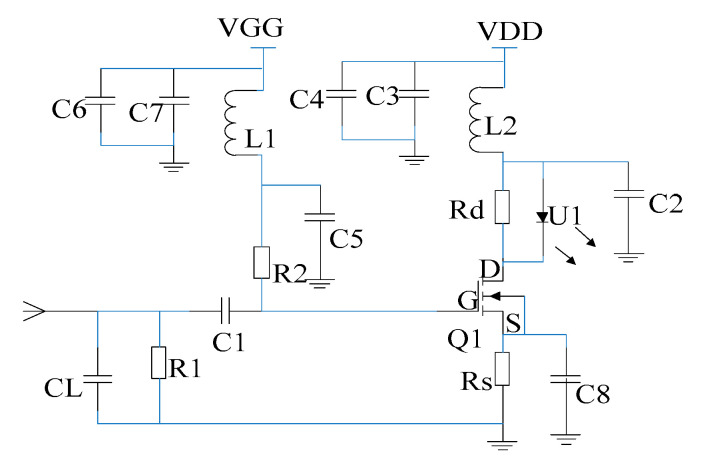
The electro-optic modulation circuit.

**Figure 3 sensors-21-08137-f003:**
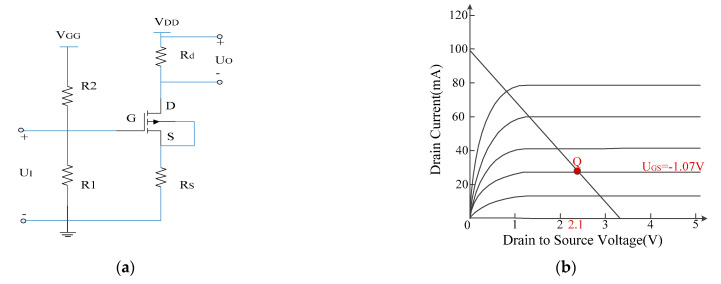
DC path of electro-optic modulation circuit (**a**) and the static operating point (**b**).

**Figure 4 sensors-21-08137-f004:**
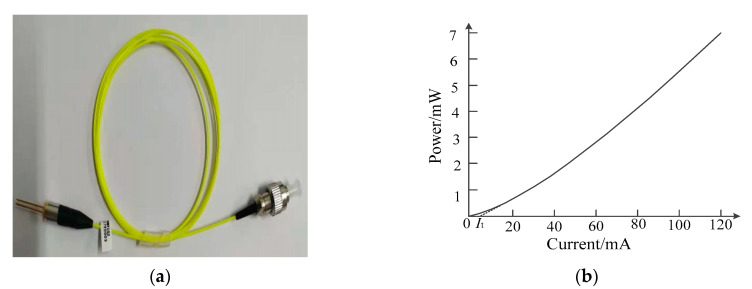
The physical diagram (**a**) and its output characteristic curve (**b**) of the semiconductor laser.

**Figure 5 sensors-21-08137-f005:**
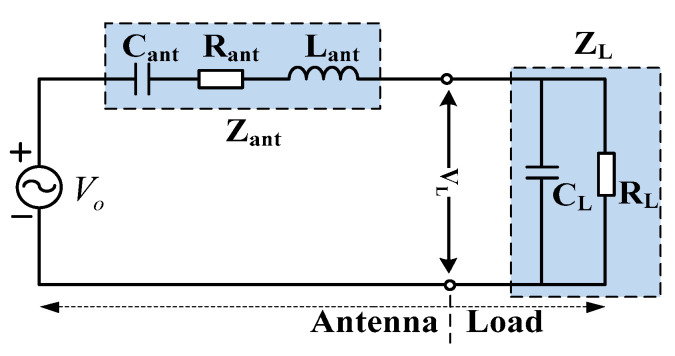
Equivalent circuit of the sensor.

**Figure 6 sensors-21-08137-f006:**
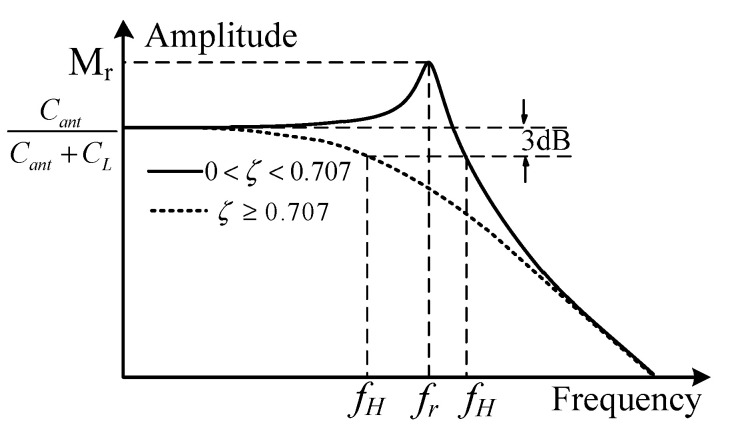
High frequency response curve.

**Figure 7 sensors-21-08137-f007:**
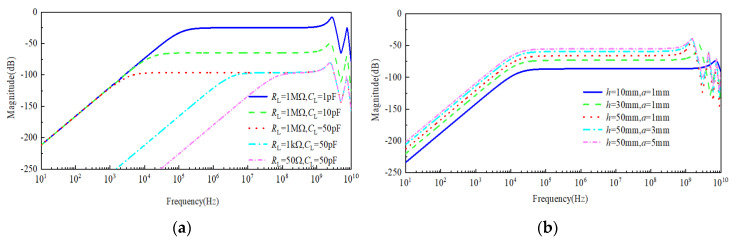
Frequency characteristic curve of the sensor under different load resistances (**a**) and different sizes of antenna (**b**).

**Figure 8 sensors-21-08137-f008:**
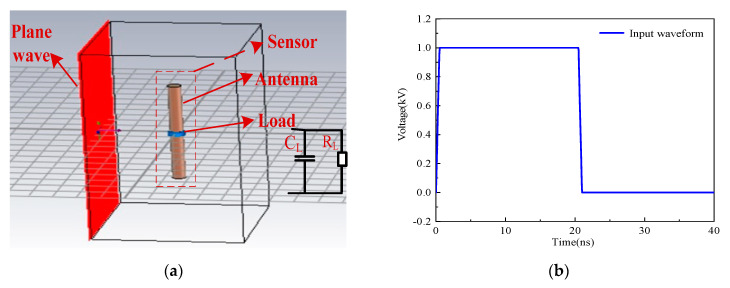
The simulation model of optical-fiber EMP field measurement system (**a**) and the input EMP field waveform (**b**).

**Figure 9 sensors-21-08137-f009:**
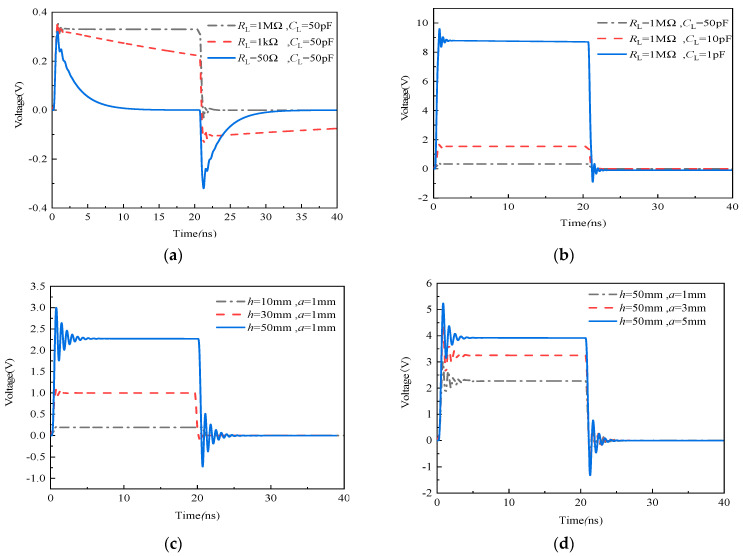
Time domain response waveform of the sensor under (**a**) different resistances *R*_L_ when *C*_L_ is 50 pF (*h* = 30 mm, *a* = 3 mm); (**b**) the different capacitance *C*_L_ when *R*_L_ is 1 MΩ (*h* = 30 mm, *a* = 3 mm); (**c**) the different length *h* of antenna when *a* is 1 mm (*R*_L_ = 1 MΩ, *C*_L_ = 10 pF); and (**d**) the different radius *a* of antenna when *h* is 50 mm (*R*_L_ = 1 MΩ, *C*_L_ = 10 pF).

**Figure 10 sensors-21-08137-f010:**
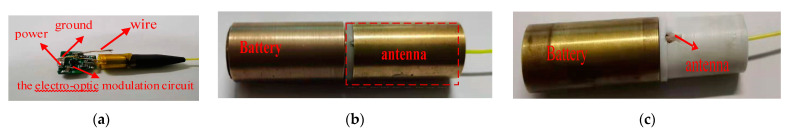
Photographs of the electro-optic modulation circuit (**a**), weak field sensor (**b**) and strong field sensor (**c**).

**Figure 11 sensors-21-08137-f011:**
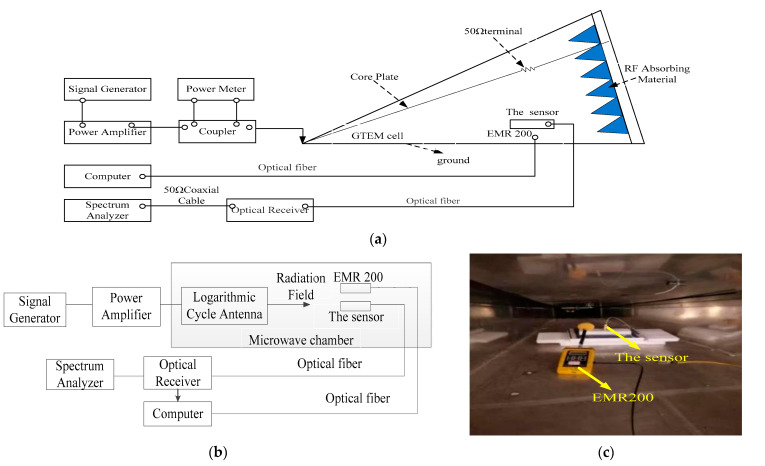
The test configuration diagram of low frequency (100 kHz–450 MHz) (**a**) and high frequency (450 MHz–1 GHz) (**b**) and the location photo of EMR200 and the sensor in GTEM cell (**c**).

**Figure 12 sensors-21-08137-f012:**
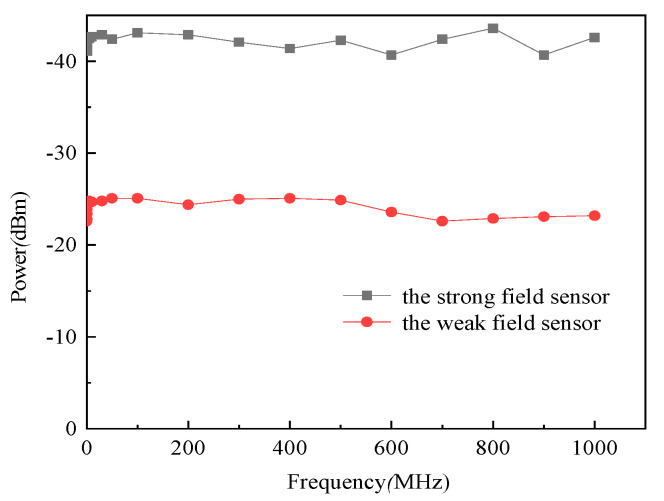
Frequency response of the sensors.

**Figure 13 sensors-21-08137-f013:**
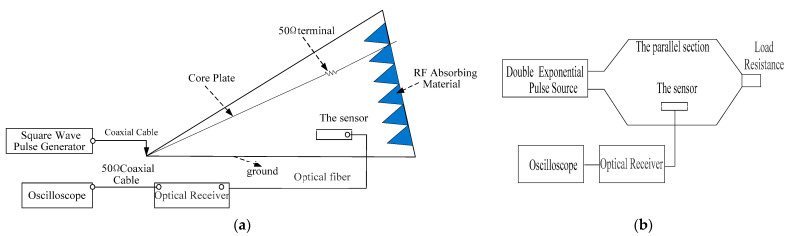
The time domain test configuration diagram of the weak field strength test (**a**) and strong field strength test (**b**).

**Figure 14 sensors-21-08137-f014:**
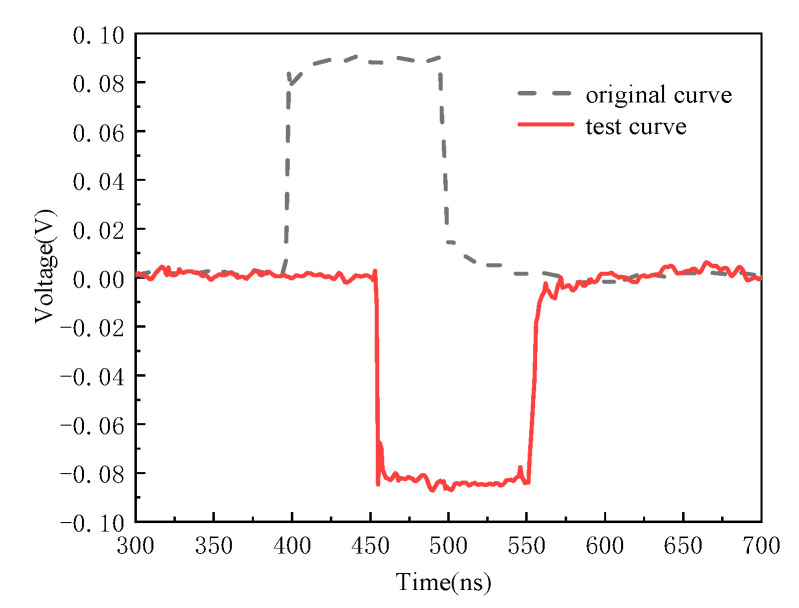
Time domain response of the sensor with the applied square wave is 8366 V/m.

**Figure 15 sensors-21-08137-f015:**
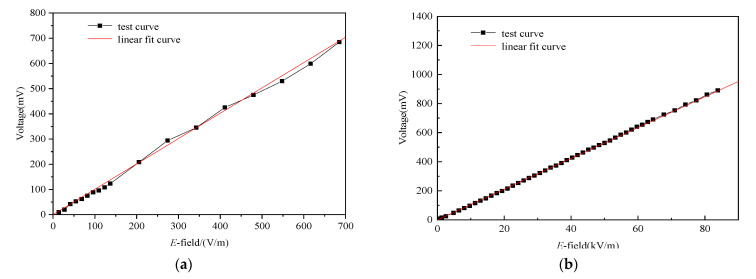
The relationship between the electric field and output voltage of the weak field sensor (**a**) and the strong field sensor (**b**).

**Table 1 sensors-21-08137-t001:** Electrical characteristics of NE72218.

PART NUMBER			NE72218		
SYMBOLS	PARAMETER AND CONDITIONS	UNITS	MIN	TYP	MAX
G_S_	Power Gain at V_DS_ = 3 V, I_D_ = 30 mA, f = 12 GHz	dB		5	
P_1dB_	Output Power at 1 dB Gain Compression Point at V_DS_ = 3 V, I_D_ = 30 mA, f = 12 GHz	dBm		15	
PN	Phase Noise at V_DS_ = 3 V, I_D_ = 30 mA, f = 11 GHz, 100 kHz offset	dBC/Hz		−110	
g_m_	Transconductance at V_DS_ = 3 V, V_GS_ = 0 V	mS	20	45	
I_DSS_	Saturated Drain Current at V_DS_ = 3 V, V_GS_ = 0 V	mA	30	60	120
V_GS(OFF)_	Grid to Source Cut Off Voltage at V_DS_ = 3 V, I_D_ = 100 μA	V	−0.5	−2	−4
I_GSO_	Grid to Source Leakage Current at V_GS_ = −5 V	μA		1	10

## Data Availability

Not applicable.

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
