# Peer review of "A Highly Sensitive and Miniature Optical Fiber Sensor for Electromagnetic Pulse Fields"

_sensors, 2021, doi:10.3390/s21238137_

Round 1

Reviewer 1 Report

  • Authors claim that “the weak field sensor 14 and the strong field sensor have flat response from 100 kHz to 1 GHz with a variation 2.3 dB and 15 2.9 dB respectively…” how the frequency range can be shifted to words higher side? What modification in design parameters and equipment’s will be required to achieve this?
  • Optical fibre is not susceptible to magnetic field…so how this electromagnetic measurement is carried out with the proposed scheme need to clarify in introduction part of the paper
  • How the optical components are correlated with electrical circuit analysis of this paper.

Reviewer 2 Report

This paper presents a finger-shaped optical fiber sensor system for the pulsed electric field measurement. The manuscript has a poor technical explanation and lacks novelty. Not clear, what type of sensor they used, and their operating principles, and sensitivity over the frequency range. The authors stated that they used a “finger-shaped fiber sensor using laser”, but never described it in the manuscript. The sensor design and experimental configuration are not appropriately described. What are the weak field sensor and the strong field sensor? The figures are not appropriate, for instance, Fig, 5, and 6. The manuscript needs extensive revisions and clear novelty with explicit technical descriptions.  

Reviewer 3 Report

The paper describes a fiber coupled laser diode which senses electric fields within certain frequency ranges and transmits information to a base station. The basic design and dynamic modeling of the system is discussed and presented for small and large signals. 

Figure 6: the labels need to be shown in English. 

Most other figures: The font size of the axes and data labels are too small to read. Also when the paper is printed in black and white it is difficult to determine which plot belongs to which scenario. Addition of data labels would be helpful to the reader. 

Round 2

Reviewer 2 Report

The authors addressed all the comments well. The manuscript is improved a lot compared to the previous version. However, still, there are many grammatical errors throughout the manuscript, and technical errors, for instance, Figure 3 caption.